# Towards Understanding of Gastric Cancer Based upon Physiological Role of Gastrin and ECL Cells

**DOI:** 10.3390/cancers12113477

**Published:** 2020-11-22

**Authors:** Helge Waldum, Patricia Mjønes

**Affiliations:** 1Department of Clinical and Molecular Medicine, Faculty of Medicine and Health Sciences, Norwegian University of Science and Technology, N-7491 Trondheim, Norway; 2Department of Pathology, St. Olav’s hospital, Trondheim University Hospital, N-7006 Trondheim, Norway; Patricia.mjones@ntnu.no

**Keywords:** cancer, ECL cell, gastric acid, gastric cancer, gastrin, histamine, neuroendocrine cell, physiology, tumour classification

## Abstract

**Simple Summary:**

Generally, we know that cancers represent genetic changes in tumour cells, but we most often do not know the causes of cancers or how they develop. Our knowledge of the regulation of gastric acid secretion is well known, with the gastric hormone gastrin maintaining gastric acidity by stimulation of the enterochromaffin-like (ECL) cell to release histamine, which subsequently augments acid secretion. Furthermore, it seems to be a general principle that stimulation of function (which, for the ECL cell, is release of histamine) in a parallel way stimulates the proliferation of the same cell. Long-term hyperstimulation of cell division predisposes to genetic changes and, thus, development of tumours. All conditions with reduced gastric acidity result in an increased risk of gastric tumours due to elevated gastrin in order to restore gastric acidity. It is probable that *Helicobacter pylori* infection (the most important cause of gastric cancer), as well as drugs inhibiting gastric acid secretion induce gastric cancer in the long-term, due to an elevation of gastrin caused by reduced gastric acidity. Gastric carcinomas have been shown to express ECL cell markers, further strengthening this relationship.

**Abstract:**

The stomach is an ideal organ to study because the gastric juice kills most of the swallowed microbes and, thus, creates rather similar milieu among individuals. Combined with a rather easy access to gastric juice, gastric physiology was among the first areas to be studied. During the last century, a rather complete understanding of the regulation of gastric acidity was obtained, establishing the central role of gastrin and the histamine producing enterochromaffin-like (ECL) cell. Similarly, the close connection between regulation of function and proliferation became evident, and, furthermore, that chronic overstimulation of a cell with the ability to proliferate, results in tumour formation. The ECL cell has long been acknowledged to give rise to neuroendocrine tumours (NETs), but not to play any role in carcinogenesis of gastric adenocarcinomas. However, when examining human gastric adenocarcinomas with the best methods presently available (immunohistochemistry with increased sensitivity and in-situ hybridization), it became clear that many of these cancers expressed neuroendocrine markers, suggesting that some of these tumours were of neuroendocrine, and more specifically, ECL cell origin. Thus, the ECL cell and its main regulator, gastrin, are central in human gastric carcinogenesis, which make new possibilities in prevention, prophylaxis, and treatment of this cancer.

## 1. Introduction

Gastric cancer has, worldwide, shown a marked reduction in prevalence during the last decades [1]. Nevertheless, gastric cancer is still an important disease being responsible for one-third of cancer deaths [1]. Furthermore, in 1995, a break in the falling prevalence was registered in young Americans [2]. *Helicobacter pylori* is the dominating cause of gastric cancer [3], and the reduction in the prevalence of gastric cancer most probably is due to a decline in *H. pylori* infection. For decades, hypoacidity has been recognized as an important factor in gastric carcinogenesis [4,5], and since the 1950s, gastritis has been associated with gastric cancer [6]. With the description of tumours in the oxyntic mucosa in rodents after long-term dosing with the first proton pump inhibitor (PPI) omeprazole [7], and the insurmountable histamine-2 (H-2) antagonist loxtidine [8], there has been concern that long-term inhibition of gastric acid secretion could also promote cancer in humans. Gastrin was early recognized to be the cause of the oxyntic mucosal tumours of enterochromaffin-like (ECL) cell origin in the rodents [9] (Table 1). The present review covers these aspects.

### 1.1. ECL Cell Properties

For one-hundred years, we have known the three principal gastric acid secretagogues: acetylcholine, gastrin, and histamine. The histamine producing ECL cell was identified by Håkanson and Owman [10] in the late 1960s, but some stuck to the mast cell as the relevant histamine producing cell until the 1980s [11]. Since then, the ECL cell has been acknowledged as the cell producing the histamine participating in the regulation of gastric acid secretion. Taking into consideration the central role of histamine in this regulation as shown by the efficient inhibition of acid secretion by H-2 antagonists [12], the control of the ECL cell function became important. Gastrin was early shown to stimulate the formation of histamine [13,14], and in isolated rat stomach [15], and isolated oxyntic mucosal cells [16,17] to augment histamine release. Thus, from a functional point of view, it was evident that the ECL cell had a gastrin receptor and by using the isolated rat stomach in combination with a fluorinated gastrin analogue in a concentration within the physiological range, we could show that the gastrin receptor was located to the ECL cell, but not to the parietal cell [18]. Quite recently, the localization of the gastrin receptor on the ECL cell and a progenitor was confirmed [19], and the closely related enterochromaffin (EC) cell was reported to serve as a reserve stem cell in the small intestine [20]. The acetylcholine analogue carbachol did not stimulate histamine release, but apparently had a direct effect on the parietal cell [21], whereas another neuro-agent, pituitary adenylate cyclase-activating polypeptide (PACAP) was a potent and efficient stimulator of histamine release [22]. In isolated ECL cells, PACAP was shown to efficiently stimulate ECL cell proliferation, even exceeding the effect of gastrin [23]. However, in vivo studies on the role of PACAP on ECL cell proliferation is missing. Somatostatin is an important physiological inhibitor of histamine release by interaction with a somatostatin-2 receptor on the ECL cell [24]. The ECL cell produces Reg protein [25], basic fibroblast growth factor (b-FGF) [26], as well as calbindin [27], and when exposed to elevated concentrations of gastrin, glycoprotein hormone α-subunit [28,29]. At least Reg protein and b-FGF can play a role during tumourigenesis. Finally, the vesicular amine transporter 2 is expressed in ECL cells [30] and may be used in the identification of this cell. The ECL cell is localized in the periphery at the base of oxyntic glands, which normally in man are found in the fundus and corpus of the stomach. However, it has become clear that oxyntic glands also occur in the antral mucosa [31], and although the ECL cell was not specifically described in oxyntic glands at this location, there is every reason to assume that these glands are complete with all cell types including the ECL cell.

### 1.2. Regulation of ECL Cell Proliferation

Generally, there is a close relationship between regulation of function and growth, which is also true for the ECL cell. Thus, gastrin is the most important regulator of ECL cell function, as well as proliferation. Hypergastrinemia induces ECL cell hyperplasia up to a certain level, at which a new equilibrium is reached [32]. Apparently, a substance from the ECL cell has a negative effect on its own proliferation. There is no threshold concentration for the trophic effect of gastrin on the ECL cell [33], and the maximal effect is reached at a concentration of a few hundred pmol/l [34,35]. There is an inverse relationship between gastric acidity and gastrin in blood [36], and the gastrin values hitherto accepted to be within the normal range are too high since, at the time of establishing gastrin immunoassays, a high proportion of asymptomatic individuals representing normal had *H. pylori-*induced gastritis, most of whom probably with reduced gastric acid secretion [37]. Every long-term hypergastrinemia, irrespective of cause, results in ECL cell hyperplasia [38,39]. Even in the normal situation, gastrin influences ECL cell growth since antrectomy resulting in gastrin reduction reduces ECL density [40]. PACAP released from the vagal nerves causes not only stimulation of ECL cell histamine release [22], but also probably mediates the trophic effect of the vagal nerves, shown in rats with unilateral vagotomy [41]. Being a neurotransmitter, PACAP´s role in gastric human physiology and pathology has not been as thoroughly investigated as gastrin. However, in mice, unilateral vagotomy was reported to suppress gastric tumourigenesis [42]. Somatostatin must, in the stomach, be regarded as a paracrine regulator influencing neighbour cells via elongations [43]. Treatment of ECL cell neuroendocrine tumours (NETs) with a somatostatin analogue may remove macroscopic tumours as well as reduce accompanying ECL cell hyperplasia [44] by interaction with a somatostatin receptor type 2 [45].

### 1.3. ECL Cell Hyperplasia

In any species, every condition with long-term hypergastrinemia results in ECL cell hyperplasia [39]. This is true for rats [7], mice [8], Japanese cotton rats [46], man with autoimmune oxyntic gastritis [47], with *H. pylori* infection affecting the oxyntic mucosa [48], with gastrinoma [49,50], and treatment with efficient inhibitors of acid secretion [38]. Even in dogs, dosing with inhibitors of acid secretion resulting in marked reduction in acid secretion, hypergastrinemia caused an increase in proliferation in the oxyntic mucosa [51]. After a variable time of hyperplasia and depending of the natural life span of the species, ECL cell derived neoplasia develops.

### 1.4. ECL Cell Neuroendocrine Tumours (NETs)

Gastrin immunoassays were developed around 1970 [52,53], and it was soon shown that patients with reduced gastric acidity had markedly elevated gastrin values [54]. Later, small tumours in the oxyntic mucosa of patients with atrophic gastritis and pernicious anaemia were described as gastric carcinoids (now called gastric NETs) [55]. The fundamental role of gastrin in the pathogenesis of gastric NETs was realized when similar tumours also occurred in patients with gastrinoma [56], causing Bordi to write a paper suggesting that these tumours were hormonally induced [57]. However, another Italian central in neuroendocrine pathology, Solcia, together with co-authors concluded that gastrin can” promote the proliferation of ECL cells but is per se apparently unable to induce ECL cell transformation” [58]. This view was supported by a recent review from the group of Robert T. Jensen [59]. Thus, in contrast to gastric NETs, gastrin has been claimed not to play any role in the development of gastric cancer. Patients with autoimmune chronic atrophic gastritis often have very high gastrin values and they are prone to develop ECL cell NETs [34,55,60,61,62,63]. Moreover, in patients with *H. pylori* gastritis, ECL cell NETs occur [64,65], but not as prevalently as in patients with autoimmune gastritis. The discrepancy between gastrin values in atrophic gastritis whether due to autoimmunity or *H. pylori* infection is most probably due to concomitant antral atrophy in the latter condition [66]. In patients with gastric NETs the inflammation has been thought to be the cause leading to ECL cell transformation, whereas hypergastrinemia then must be responsible for the ECL proliferation and perhaps should make these cells more prone for tumour development.

However, in patients with gastrinoma, there is no inflammation in the oxyntic mucosa. The transformation of ECL cells into tumour cells has been explained by the genetic defect since they occur mainly in patients with gastrinoma as part of multiple endocrine neoplasia type I (MEN-I) [56,67]. These patients have an increased frequency of tumours originating from many different endocrine cells. Nevertheless, ECL cell NETs also occur, but more seldom, in gastrinoma patients without MEN-1 [68,69] indicating that hypergastrinemia is enough to induce such tumours. The ECL cells in sporadic gastrinoma patients also show expression of the alpha subunit of human chorionic gonadotropin [33]. Moreover, gastric NETs have been described in patients with elevated gastrin due to long-term treatment with inhibitors of gastric acid secretion, particularly proton pump inhibitors [70,71,72,73]. Therefore, it seems obvious that hypergastrinemia itself is enough to induce gastric NETs in man, as has been shown for rodents after long-term drug induced hypoacidity [7,8] or due to partial corpectomy [74]. Moreover, in man, surgery-induced hypergastrinemia leads to gastric NETs [75].

### 1.5. ECL Cell NETs and Gastric Carcinomas

It has long been accepted that the ECL cell may develop into neuroendocrine carcinomas defined as type 3, according to Rindi et al. [76]. These tumours occur in patients without hypergastrinemia but taking into consideration that the ECL cell has a growth-stimulating gastrin receptor, gastrin could nevertheless play a role in tumour development. However, such tumours probably develop by chance-mutation in a gene central in ECL cell growth. Gastric cancers are not seldom detected in patients with ECL cell NETs secondary to autoimmune gastritis [77,78]. Moreover, in a systematic literature search on the co-occurrence of NETs, and adenocarcinomas in the same segment of the gastrointestinal tract, a highly significant association was found [79]. By applying immunohistochemistry with improved sensitivity, we could show that nearly all (seven of eight) carcinomas removed from patients with marked hypergastrinemia expressed neuroendocrine markers [80], suggesting that they originated from neuroendocrine cells. Therefore, the gastric cancers in patients with pernicious anaemia, until now classified as adenocarcinomas, may be neuroendocrine carcinomas developed from ECL cells. In general, when using immunohistochemistry with increased sensitivity we demonstrated neuroendocrine differentiation in tumour cells in a significant proportion of gastric cancers [81]. By immunoelectron microscopy, we also demonstrated chromogranin A positive granules in gastric cancer cells [82]. We have previously focused on the classification of gastric carcinomas based on non-specific histochemical methods, not taking notice of much more specific and sensitive methods, such as immunohistochemistry and in-situ hybridization [83]. Thus, the unspecific periodic acid-Schiff (PAS) stain method has been accepted as a proof of mucin. PAS positive tumour cells were therefore regarded to be of exocrine cell origin enabling classification of tumours as adenocarcinomas, even without glandular growth pattern. However, we could not detect mucin expression, either by immunohistochemistry or in-situ hybridization, in contrast to neuroendocrine expression in many gastric PAS positive carcinomas of particularly the diffuse type, according to Lauren [84]. Neuroendocrine expression was especially marked in the signet ring cell subtype [85], a finding also reported by others [86,87,88]. It is therefore possible that the ECL cell gives rise to an important proportion of gastric carcinomas. We have also described a patient with pernicious anaemia who developed a gastric NET, which was removed endoscopically but re-occurred in lymph nodes, which were removed surgically. After some years, the patients died of a highly malignant neuroendocrine carcinoma demonstrating that gastric ECL cells have a malignant potential [89]. Furthermore, in a Spanish family with a missense mutation in one of the genes coding for the proton pump, not only gastric NETs, but also a carcinoma were described [90], although we later reclassified the latter tumour as a neuroendocrine carcinoma [91]. Reclassification of adenocarcinomas to neuroendocrine tumours was suggested more than 40 years ago [92], and an adenocarcinoma producing neuron-specific enolase [93], a neuroendocrine marker more specific than hitherto appreciated [94], has been reported.

The ECL cell may play an important role in gastric carcinogenesis also indirectly by the release of Reg protein, which has been shown to stimulate proliferation of gastric cells and differentiation along parietal and chief cell lineages [95], and thus mediate the general trophic effect of gastrin on oxyntic mucosa [25]. An increase in Reg protein release due to ECL cell hyperplasia would cause a chronic growth stimulation of the stem cell which in long-term would be expected to increase the risk of cancer development, presumably of the intestinal type, which also has been reported [96]. In this connection, we will recall that hypergastrinemia presently is the most probable mechanism for the carcinogenic effect of *H. pylori* gastritis [97]. The carcinogenic effect of *H. pylori* gastritis seems not to be the infection directly, but indirectly by the secondary atrophy of the oxyntic mucosa [98], which necessarily will lead to hypergastrinemia and ECL cell hyperplasia. The hyperplastic ECL cell may develop further into tumour cells leading to ECL cell NETs or further to neuroendocrine carcinomas and carcinomas hitherto classified as adenocarcinomas, and then, mainly of diffuse type according to Lauren. Alternatively, by stimulation of the stem cell via Reg protein, the ECL will predispose to adenocarcinoma, mainly of intestinal type.

The normal ECL cell as well as ECL cell in NETs produce b-FGF [26]. The fibrosis, which is a central feature of diffuse gastric carcinomas as exemplified by linitis plastica, may be due to release of b-FGF from ECL cell derived tumour cells. In fact, overexpression of b-FGF mRNA was reported particularly in carcinomas of scirrhous type [99], which ordinarily are heavily fibrotic. Moreover, there is a case report of a patient with scirrhous type gastric cancer who had aggressive fibroses in the head and neck with scattered b-FGF positive gastric cancer cells [100]. This cancer could be an example of highly malignant tumour derived from the ECL cell.

### 1.6. Mixed Tumours with Neuroendocrine and Adenocarcinoma Components

There are many descriptions of reclassification of gastric carcinomas initially classified as adenocarcinomas and later reclassified as neuroendocrine carcinomas [101,102,103], demonstrating difficulties in the distinction between these two entities in man, such as in rodents [104].

### 1.7. The Carcinogenic Effect of Hypoacidity, Whether Caused by Oxyntic Gastritis, Due to “Autoimmunity” or H. pylori, as Well as Inhibitors of Gastric Acid Secretion, is Mediated by Gastrin

Hypoacidity was the first condition recognized to play a role in gastric carcinogenesis [5] followed by gastritis a few years later [6] when leading to oxyntic atrophy [98], hypoacidity, and, thus, hypergastrinemia. With the central role of gastrin in gastric carcinogenesis, it was therefore to be expected that long-term treatment with efficient inhibitors of gastric acid secretion necessarily would predispose to gastric neoplasia. The target cell of gastrin, the ECL cell, develops into neoplasia when exposed to long-term hypergastrinemia in all species examined adequately. When examining gastric carcinomas with the most sensitive and specific methods available, ECL cell differentiation is revealed in many of the tumours until now classified as adenocarcinomas [83]. Moreover, the practice of dismissing neuroendocrine differentiation in cancer cells when found only in a few of them [105,106,107], seems from a biological point of view very odd. Furthermore, there are many reports of gastric tumours composed of a NET, as well as an adenocarcinoma component [108,109,110], or neuroendocrine carcinoma and adenocarcinoma [111] showing the close connections between these entities. Gastrin receptors have also been described on gastric carcinoma cells [112,113]. Gastric cancer may also be due to infection with Epstein Barr virus [114]. Interestingly, the genetic changes in carcinomas due to Epstein Barr virus and *H. pylori* do not share similarities [115], suggesting different mechanisms in the carcinogenic process. There is one publication describing different genetic patterns between gastric carcinoma of diffuse and intestinal type [116], but this has not been confirmed. Finally, there are two known hereditary conditions predisposing to gastric cancer, mutation of the CDH 1 gene coding for E-cadherin predisposing to diffuse gastric cancer at young age [117] and mutation of the ATP4A gene coding for one of the genes of the proton pump leading to ECL cell derived tumours of variable malignancy [90,91] (Table 2).

Chronic inflammation is believed to be an important factor in carcinogenesis, including gastric cancer secondary to gastritis [118,119]. However, *H. pylori* gastritis does not seem to have a direct carcinogenic effect, since the risk is eliminated when *H. pylori* is eradicated at an early phase before having caused oxyntic atrophy [120]. Those having developed oxyntic atrophy maintain risk of gastric cancer years after having lost the infection, due to lack of gastric acidity or *H. pylori* eradication [121,122]. In general, to our knowledge, a bacterium has hitherto not been shown to have a direct carcinogenic effect. On the other hand, the gastric lymphoma secondary to *H. pylori* infection is cured by *H. pylori* eradication [123]. In this overview, we have not dealt separately with the role of food in gastric carcinogenesis. The reason why is that we doubt the importance of food as a separate factor in this process. The food, together with water, may, of course, play a role in the carcinogenesis in the stomach by carrying microbes like *Helicobacter pylori* and Epstein Barr virus and probably others to the stomach. Therefore, exposure of food to enough high temperature for necessary time will probably have a preventive effect.

## 2. Clinical Implications

Accepting a central role of the ECL cell in gastric carcinogenesis has clinical consequences as it also implies a central role of gastrin, the main regulator of the ECL cell. Chromogranin A (CgA) is a sensitive marker for ECL cell mass [124] and ECL cell function, since infusion of a somatostatin analogue reduces chromogranin almost immediately, and long before any reduction in ECL cell mass [125]. Together with the main regulator of the ECL cell, gastrin, CgA may be used in the diagnoses of autoimmune chronic atrophic gastritis [126], which is the main factor in gastric cancer whether induced by *H. pylori* [98] or autoimmunity [61]. Moreover, CgA reflects 24-h gastrin better than fasting morning gastrin in patients treated with proton pump inhibitor once daily [124]. The new group of inhibitors of gastric acid secretion, the potassium-competitive acid blockers represented by vonoprazan seem to be more efficient in inhibiting acid secretion, but as was to be expected, similarly induce more hypergastrinemia [127]. Thus, vonoprazan should be reserved for only short-term treatment where profound acid inhibition is required, such as for *H. pylori* eradication or acute upper gastrointestinal bleeding. Both the gastrin antagonist netazepide [128] and the somatostatin analogue octreotide [44] make gastric ECL cell NETs disappear, and it would be interesting to treat gastric carcinomas expressing both somatostatin 2 and gastrin receptors with a combination of these two agents, each having very little side effects.

Furthermore, *H. pylori* should be eradicated as early as possible, before irreversible changes have occurred in the oxyntic mucosa. This would probably be best achieved by testing H. pylori serology in young adults since H. pylori infection mainly occurs in children [129], and treating persons testing positive. In those with advanced oxyntic atrophy shown by endoscopy with biopsies, or markers of oxyntic atrophy, such as elevated gastrin or low pepsinogen I [130], or demonstration of anacidity in combination with negative *H. pylori* serology, the logical attitude would be to prevent continuous gastrin stimulation by using the gastrin antagonist netazepide [128]. Netazepide has very few, if any, side effects [131], but is unfortunately not yet available. Alternatively, or complementary, those patients should be followed by regular upper endoscopy. Furthermore, *H. pylori* infection increases serum gastrin during PPI treatment, as shown by a fall when eradicating the bacterium and continuation of the PPI treatment [132]. A proposal for treatment of *Helicobacter pylori* infection is outlined in Table 3.

Finally, the reason why the efficient inhibitors of acid secretion, such as PPIs, and even the more efficient potassium-competitive acid blocker [133] are superior to less efficient drugs in the treatment of acid-related diseases is their ability to reduce gastric acid more, and for a longer time, which necessarily will result in more pronounced hypergastrinemia [127]. Thus, increase in fasting serum gastrin when treated with PPI once daily was inversely correlated to time of 24 h pH below 4.0, a level regarded as a treatment goal [134]. Long-term treatment of PPIs has been reported to lead to gastric carcinoma [135], and the carcinogenic effect of lack of function of the proton pump is also shown in the family with ATP4A mutation [90]. The tumourigenic effect of hypergastrinemia was first shown in rodents in the middle of the eighties [7,8] and later confirmed in cotton rats [46,136]. The INS-GAS mice model (transgenic mice where the gastrin gene is under control of the insulin promotor) demonstrates the importance of gastrin in gastric tumourigenesis, as well as the additive carcinogenic effect of gastrin and *H. pylori* infection [137]. We are pleased that we no longer are alone with the present views, since a Chinese group, one year ago in a review, seemed to share most of them [138]. We will also add that metastases of slow-proliferating cancer cells, such as neuroendocrine cells, may explain the apparent dormancy of cancer cells giving rise to very late cancer recurrence [139]. The central role of the ECL cell in gastric physiology and pathology [140] is depicted in Figure 1.

Finally, it is important to understand that hormonal carcinogenesis does not dismiss the importance of gene mutations in carcinogenesis. Mutations and other genetic changes are, of course, central in the carcinogenic process. However, hormones and other growth factors predispose to cancer by their stimulation of proliferation. Every cell division implies a risk of mutation. Thus, mitogens are also mutagens [141]. Hormones are complete carcinogens also in man, as demonstrated by the cancers in vagina/vulva of girls, born to women having been treated with oestrogen during pregnancy, with the purpose of reducing risk of abortion [142]. Only about 3% of gastric cancers have an inherited component, leaving most genetic changes in gastric cancers to acquired causes [143].

## 3. Conclusions

The role of gastrin and its target cell, the ECL cell, in gastric carcinogenesis seems, for us now, to be overwhelming. Together with sex hormones, oestrogens causing breast cancer and androgens cancer of the prostate, the role of gastrin in gastric cancer are examples of hormonal carcinogenesis. Furthermore, it shows the central role of physiology in tumourigenesis, and together with *H. pylori* gastritis, has a key role in the pathogenesis of diseases in the upper gastrointestinal tract. Clinical gastroenterology will certainly profit from this new understanding and improve prophylaxis and treatment.

## Figures and Tables

**Figure 1 cancers-12-03477-f001:**
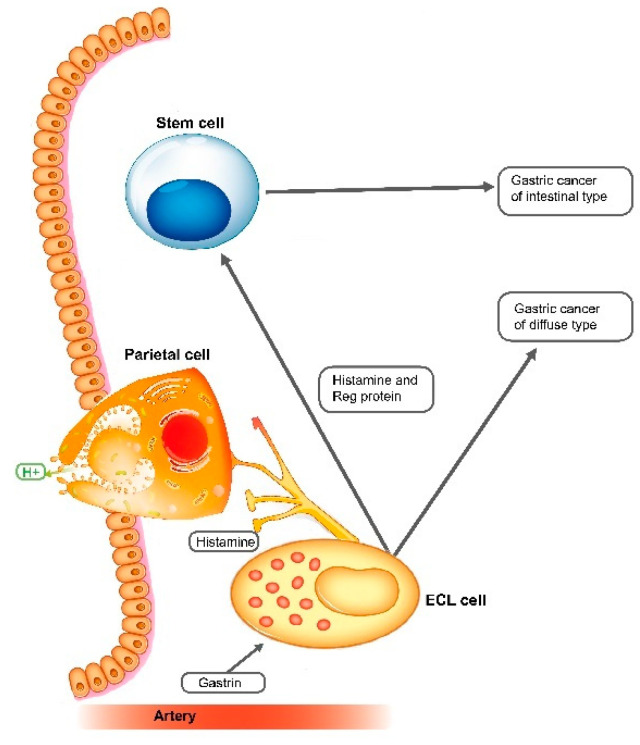
The central role of gastrin and the enterochromaffin-like (ECL) cell in the regulation of gastric acid secretion and in gastric carcinogenesis (reproduced by permission from the publisher, reference [139]).

**Table 1 cancers-12-03477-t001:** Conditions acknowledged, predisposing to gastric cancer at different points of time.

Condition	Time
Hypoacidity	Forties
Gastritis	Fifties
*Helicobacter pylori*	Early nineties
Gastrin	Two thousand

(Gastrin is the common factor for the first three conditions).

**Table 2 cancers-12-03477-t002:** Causes of gastric cancer.

Causes of Gastric Cancer	Gastrin Driven	Probably not Gastrin Driven
*Helicobacter pylori*	x	
Autoimmune gastritis	x	
Drugs inhibiting gastric acid secretion	x	
Genetic; ATP4 mutation [90]	x	
Genetic: Hereditary diffuse gastric cancer, CDH 1 mutation [117]		x
Virus: Epstein Barr virus [114]		x

**Table 3 cancers-12-03477-t003:** Treatment of *Helicobacter pylori* related gastritis to prevent gastric cancer.

Test for *Helicobacter Pylori* (Hp) Antibodies	Treatment Options	
Young—about 20 years	Hp eradication	
Before starting long-term PPI treatment	Hp eradication	
Those with oxyntic atrophy without Hp		Upper endoscopy/gastrin antagonist

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
