# Peer review of "Towards Understanding of Gastric Cancer Based upon Physiological Role of Gastrin and ECL Cells"

_cancers, 2020, doi:10.3390/cancers12113477_

Round 1
Reviewer 1 Report
The authors treat exhaustively the focus of the review that I consider innovative.
Just two notes:
-ECL specify the full name, not just the acronym.
- Insert a final illustrative scheme to allow a clearer and more immediate understanding of the topic described in the review.
Author Response
Thank you for a positive report. We have given the full name (Enterochromaffin like) in the title heading.
We have added a figure
Reviewer 2 Report
Newsworthy and complete review studying the role played by ECL cells and gastrin in the risk of gastric carcinoma’s onset. Novel is the perspective of the analysis of the centrality of gastric physiology in tumorigenesis, the understanding of which represents a further tool for an adequate prevention and early treatment of upper GI pathologies. The analysis of the literature is detailed and specific, the study of the pathophysiological functions of ECL cells implicated in neuroendocrine carcinoma and gastric carcinoma development, both in the animal models and in their possible clinical implications, describing also pharmacological features, is interesting and opens up new views for future dedicated studies.
Continuous efforts towards improvements in cancer prevention and early diagnosis are crucial. This review is detailed and offers a new view in studying gastric carcinogenesis and related physiological implications.
Author Response
Thank you for a positive evaluation
Reviewer 3 Report
In the present review entitled “Towards understanding of gastric cancer based upon physiological knowledge of the central position of gastrin and the ECL cell”, the authors Waldum HL and Mjones P emphasized that many of gastric cancers are histamine producing ECL cell origin. Their speculation appears to be based on many physiological knowledges from gastric cancer patients and limited pathological report (reference #76 by Qvigstad G et al.). In the previous pathological report, eight gastric adenocarcinomas found in patient with severe hypergastrinemia and/or pernicious anemia were examined with immunohistochemical technique and scattered neuroendocrine tumor cells were found in seven of the carcinomas. This pathological report should not be sufficient to conclude many gastric cancers to be ECL cell origin. In addition, recent molecular researches including whole genome sequencing approaches demonstrated that gastric cancers harbor many gene mutations in tumor suppressors, e.g., TP53 and BRCA2 and cell proliferations, e.g., WNT and PIK3CA, and long-term hyperstimulation of cell proliferation by hypergastrinemia can not effectively explain to predispose to these genetic changes. Therefore, it is concerned that the present review might add confusion to the treatment/prevention strategy of gastric cancers.
Author Response
We appreciate the review reports from referees.
Reviewer 3 gave us the lowest possible evaluation of 4 of the 5 points, but gave a medium score for references. The latter is surprisingly since he seemed to mean that our conclusions in pathology was based on only one study.
This referee claims that our review lacked studies in pathology only briefly discussing one study. In fact, studies related to gastric neoplasia pathology are covered in the following references:
Rodents; ref 7, 8, 44, 71, 96.
Humans; 24, 27, 31,32, 36, 53, 54, 55, 56, 57, 58, 59, 60, 61, 62, 64, 65,66, 67, 68, 69,70, 72, 73, 74, 75, 76,77,78,79,80,81,82,83, 84,85,86,89,90,91, 92, 93,94, 95,97,98,99,100,101,102,103,104,105,106,107,108,109,110,111,112, 113, 114,115,116,118,119,121,124,128,131.
Furthermore, gastrin and gastric cancer are discussed separately in papers 35, 37, 46, 47, 48, 52. Taking into consideration that the ECL cell is the only cell without ambiguity having a gastrin receptor, gastrin involvement also must indirectly has influence for the role of the ECL cell.
I could not see that referee 3 denies an important role of gastrin in gastric carcinogenesis.
Referee 3 does not otherwise come with many other arguments except asking for genetic studies mentioning mutations that could not be induced by gastrin. I am very pleased with this note since mutations are a fundamental common factor in all tumours. Stimulation of proliferation, mitogens are also mutagens. Chronic overstimulation of proliferation is known to cause neoplasia in animals and humans (for instance the cancer of vagina in young girls born of mothers after having been treated with oestrogen in the hope of reducing the risk of abortion).
We have added two more references related to pathology ( one on immunohistochemistry with high sensitivity and the other on immuno electron microscopy.
Furthermore, since the referee seems to mix genetic alterations in cancer cells with the cause of these changes, we have added a short paragraph before the conclusion. Hopefully, this will clarify the apparent discrepancy in view.
Reviewer 4 Report
I reviewed the comments of reviewers and author's reply.
It is widely known that general gastric cancer, not carcinoid tumors, is caused by several factors.
Therefore, I think that the authors should emphasize that roles of gastrin and ECL are one of carcinogenesis of gastric cancer.
Author Response
I will change the phrase to: Whereas gastrin is accepted as causing gastric NETs (carcinoids), its role in gastric carcinogenesis has been controversial.
Round 2
Reviewer 3 Report
- I agree that carcinomas developed from ECL cells, so called carcinoids, are promoted by hypergastrinemia.
- A sentence “There is also accumulating evidence that a proportion of gastric carcinomas of the diffuse type is derived from the ECL cell”, can be found in ref. 37 (this review was from one of the present authors). A fundamental question is whether is this widely accepted or not.
- The several references which are mentioned in the authors’ response, e.g., #46 and #47 focused on ECL cell hyperplasia or fundic argyrophil cell densities, but not on gastric cancers.
- I do not know the types of gene mutations found in typical carcinoids, but they do not probably resemble those found in typical gastric cancers.
- In conclusion, it is concerned once again that the present review might add confusion to the treatment/prevention strategy of gastric cancers.
Author Response
Thank you for the new evaluation.
I will start with the conclusion that our manuscript will add confusion for the treatment/prevention strategy of gastric cancer.
This statement is rather odd taking into consideration that there has been virtually no progress within the field of gastric cancer treatment for decades in spite of a major methodological progress like upper endoscopy and increased knowledge by the description of Helicobacter pylori as the major gastric carcinogen. Perhaps lack of openness for other ideas may have prevented progress? Concerning the mechanism for the carcinogenic effect of Helicobacter pylori, it has not been found although heavily sought for 30 years. A bacterium has hitherto not been shown to be direct carcinogenic, and it has been clearly shown that the carcinogenic effect is related to oxyntic atrophy which necessarily lead to hypergastrinemia. Moreover, proton pump inhibitors predispose to gastric cancer rather early in patients having shortly before been eradicated for Helicobacter pylori (Cheung et GUT 2018) indicating an additive effect.
The enhancement of neuroendocrine tumours by PPI administration to Helicobacter pylori infected Mongolian gerbils also is in line with a common mechanism for the tumours (Tsukamoto. Asian Pas. J Cancer Prev. 2011 Similarly, the development of gastric cancer many years after Helicobacter pylori eradication (Take. J Gastroenterol 2020)clearly indicates that Helicobacter pylori per se is not carcinogenic but that the carcinogenesis is related to the oxyntic atrophy. Since every condition with long-term hypergastrinemia is accompanied by increased risk of gastric cancer in every species examined including man, it is difficult to understand that our manuscript should be so controversial.
Then, the role of the ECL cell which has the gastrin receptor. First a general statement: During the process of malignancy, the cell loses specific traits. Therefore, methods with the highest sensitivity must be used to detect the markers. It is strange that the referee did not comment on our study demonstrating the gastrin receptor on the gastric cancer cells (Mjønes et al).
I will also mention two recent studies with relevance to the gastrin receptor as well as cancer. Sheng et al. (CMGH 2020) showed that the gastrin receptor was expressed on ECL cells and some progenitor cells in the isthmus, and an even more recent study by Sei et al (AMJ Physiol; Gastrointest, Liver Physiol. 2020) where enteroendocrine cells in the small intestine were described to give rise to reserve stem cells. It should not be necessary to add that the EC cells in the small intestine are closely related to the ECL cells.
The aspects written above will be included in the next version.
I will comment on the remarks from the reviewer point per point.
We agree upon the first point on gastrin and gastric carcinoids (NETs).
We will stick to the second phrase which is substantiated by what is written above and will be included in the manuscript.
My question is: What progress will be made if all that is written is widely accepted?
More references will be included.
The referee is very concerned about genetic changes although he could not come up with any references. There seems to be no pattern of mutations in gastric cancers although those induced Epstein-Barr virus were reported to be a distinct entity ( Curr Top Microbiol Immunol 2017; 400; 277-304). An older paper ( IARC Sci Publ 2004; 157: 327-349.) described different genetic patterns between intestinal and diffuse types of cancer. I do not know of any study describing genetic changes in gastric NETs which perhaps is not surprising taking into consideration that few such changes would be expected, since these tumour cells morphologically are very similar to normal ECL cells. However, there are changes as these cells like hyperplastic cells for instance express alfa-glyco-hormones (Bordi). I will add a paragraph on the genetic changes in gastric NETs and cancers.
I will conclude that our paper is based on physiology, pathophysiology and solid pathology where we have applied the most sensitive and specific methods (immunohistochemistry with signal amplification, in-situ hybridization and immuno- electron microscopy) both on human tumours as well as animal ones.